# The Effect of miR-140-5p with HDAC4 towards Growth and Differentiation Signaling of Chondrocytes in Thiram-Induced Tibial Dyschondroplasia

**DOI:** 10.3390/ijms241310975

**Published:** 2023-06-30

**Authors:** Wangyuan Yao, Muhammad Fakhar-e-Alam Kulyar, Yanmei Ding, Haitao Du, Jiajia Hong, Kyein San Loon, Shah Nawaz, Jiakui Li

**Affiliations:** 1College of Veterinary Medicine, Huazhong Agricultural University, Wuhan 430070, China; wangyuay@ucr.edu (W.Y.); fakharealam786@hotmail.com (M.F.-e.-A.K.); dingyanmei@webmail.hzau.edu.cn (Y.D.); duhaitao0920@163.com (H.D.); hong2_2@163.com (J.H.); sannu1123@gmail.com (K.S.L.);; 2Department of Microbiology and Plant Pathology, University of California, Riverside, CA 92521, USA; 3Department of Animal Nutrition and Feed Science, College of Animal Science and Technology, Huazhong Agricultural University, Wuhan 430070, China

**Keywords:** microRNA-140-5p, growth plate, histone deacetylase, chondrocyte hypertrophy, endochondral ossification

## Abstract

There is evidence to suggest that microRNA-140-5p (miR-140), which acts as a suppressor, is often elevated and has a role in various malignancies. Nevertheless, neither the function nor the mechanisms in chondrocytes linked with bone disorders, e.g., tibial dyschondroplasia (TD), have been satisfactorily established. The purpose of this study was to look into the role of microRNA-140-5p (miR-140) and its interaction with HDAC4 in chondrocytes, as well as the implications for tibial dyschondroplasia (TD), with a particular focus on the relationship between low miR-140 expression and poor pathologic characteristics, as well as its physiological effects on chondrocyte growth, differentiation, and chondrodysplasia. In this investigation, we discovered that TD had a reduced expression level of the miR-140. There was a correlation between low miR-140 expression, poor pathologic characteristics, and the short overall survival of chondrocytes. Our findings show an aberrant reduction in miR-140 expression, and HDAC4 overexpression caused disengagement in resting and proliferation zones. This further resulted in uncontrolled cell proliferation, differentiation, and chondrodysplasia. Mechanistically, HDAC4 inhibited the downstream transcription factors MEF2C and Runx2 and interacted with Col-Ⅱ, Col-X, and COMP. However, miR-140 binding to the 3′-UTR of HDAC4 resulted in the growth and differentiation of chondrocytes. Moreover, the expression of HDAC4 through LMK-235 was significantly decreased, and the expression was significantly increased under ITSA-1, referring to a positive feedback circuit of miR-140 and HDAC4 for endochondral bone ossification. Furthermore, as a prospective treatment, the flavonoids of Rhizoma drynariae (TFRD) therapy increased the expression of miR-140. Compared to the TD group, TFRD treatment increased the expression of growth-promoting and chondrocyte differentiation markers, implying that TFRD can promote chondrocyte proliferation and differentiation in the tibial growth plate. Hence, directing this circuit may represent a promising target for chondrocyte-related bone disorders and all associated pathological bone conditions.

## 1. Introduction

Tibial dyschondroplasia (TD) is the most intractable cartilage disease, characterized by the degeneration of tibial cartilage, the thickening of the growth plate, obstruction of vascular invasion, and the formation of opaque cartilage plugs [1]. The pathogenesis of TD is complex and diverse, but the uncontrolled growth, differentiation, and apoptosis of growth plate chondrocytes are recognized as the primary evidence [2,3,4]. Although significant coupling between the regulation of chondrocyte growth, terminal differentiation, and the recruitment of growth plate blood vessels has been demonstrated [5,6,7], the mechanisms underlying precise the coordination of proliferation, differentiation, formation, and the invasion of blood vessels remain unknown in the growth plate [8,9].

MicroRNA (miRNA), a short endogenous oligonucleotide with approximately 22 nucleotides, participates in the post-transcriptional regulation of cell proliferation, differentiation, apoptosis, metabolism, and other functions by binding to the 3′ untranslated region (3′-UTR) of target mRNA. Cartilage development depends on the proliferation, differentiation, and calcification of chondrocytes and the strict regulation of critical factors [10,11]. One of these factors is miRNA, which plays a pivotal role in cartilage differentiation, formation, and cartilage lesions. Additionally, mature miRNA is necessary for normal bone development [12,13]. Lin et al. revealed for the first time that miR-199 has an inhibitory effect on early chondrogenesis during cartilage development through cell chip technology, which is involved in bone regulation [14]. Subsequently, some important miRNAs involved in cartilage formation and chondrocyte development, such as miR-138, miR-218, miR-340-5p, miR-27b-3p, miR-127-5p, miR-34a, miR-4784, miR-206, miR-31, and miR-93, etc., many researchers have a strong interest in the research of the functions and mechanisms of different miRNAs in different animals [15,16,17,18,19,20]. Rasheed’s study of human chondrocytes found that miR-125b-5p participated in regulation by targeting the TRAF6/MAPKs/NF-κB signaling pathway [21]. miR-27b-3p and miR-193b-3p were revealed to regulate the growth and development of human chondrocytes by inhibiting the expression of HIPK2 and matrix metalloproteinase 19 [22,23]. In addition, recent studies have reported that miR-31, miR-590-5p, miR-98, miR-181a/b-1, etc., regulate human chondrocytes [24,25,26]. Our research team discovered many differentially expressed miRNAs related to cartilage development in the tibial growth plate of TD broilers through high-throughput sequencing technology. Among them, the expression of miR-140-5p in the tibial growth plate of TD broilers significantly decreased [27].

miR-140 was considered to play an essential role in mouse cartilage homeostasis. Li et al. confirmed that the potential role of miR-140-5p helped regulate mouse mandibular condylar cartilage homeostasis and was used as a new prognostic factor for mice, e.g., osteoarthritis (OA) [28]. Another study showed that the expression levels of miR-140 and BMP-2 could reveal the changes in synovial fluid of OA patients before and after arthroscopic debridement [29]. In addition, miR-140 is crucial to cartilage homeostasis during human bone growth and development. Recently, Giedre et al. discovered a novel morphological seed area mutation of the chondrocyte-specific and super enhancer-related miR-140 gene encoding microRNA-140 (miR-140) in a novel autosomal dominant human cartilage dysplasia. Unlike miR-140-null mice, mice with identical single nucleotide mutations had skeletal deformities [30]. However, we found that it has a similar regulatory role in poultry growth plate chondrocytes, but its function and mechanism in growth plate chondrocytes are unclear.

Histone deacetylase (HDAC) is a protease that modifies the structure of chromosomes and regulates gene expression, thereby participating in cell growth, differentiation, and apoptosis [31]. At present, up to 18 HDACs have been found in animals. According to the structure and function differences, histone deacetylases are divided into three categories. Class I HDACs (HDAC1, 2, 3, and 8) are widely expressed in organisms, primarily in the catalytic domain. Class II HDACs (HDAC4, 5, 7, and 9) show strict tissue expression specificity. It contains an N-terminal extension that links to specific transcription factors and confers different signal transduction systems for connecting the genome to the extracellular environment. Class III HDACs have distinct evolutionary properties, including anti-aging enzymes. Studies have confirmed that HADC4 is closely related to the hypertrophy of chondrocytes [32,33]. Vega et al. found that HDAC4 delays chondrocyte hypertrophy and prevents endochondral osteogenesis by binding and inhibiting the activity of the pro-chondrocyte hypertrophy transcription factor Runx2. This study achieved ectopic bone formation by knocking out HDAC4 in mouse chondrocytes, which was distinctly characterized by the hypertrophy of chondrocytes [34]. Other studies showed that HDAC4 inhibited chondrocyte hypertrophy and endochondral bone development by directly interacting with MEF2C, inhibiting its expression and function [35]. The variables mentioned above form a complex regulatory framework that ensures appropriate skeletal growth; however, HDAC4 is considered the central point of the regulatory network [36]. It has been considered a potential target gene miRNA-140 for the tight regulation of post-transcriptional expression [37]. Although these studies have shown that HDAC4 plays a vital role in the growth and development of cartilage, the targeted regulatory effect on HDAC4 and the biological mechanism remains unclear.

Growth plate chondrocytes were utilized in this work as a model to understand the role of miR-140-5p in chondrocyte growth and differentiation in vitro. It was also utilized to analyze how Rhizoma drynariae affects miRNA expression in vivo to understand the role of miR-140-5p in the pathogenesis of TD.

## 2. Results

### 2.1. Tibial Dyschondroplasia May Be Due to the Inability of Chondrocytes to Transition between the Resting Zone and the Proliferative Zone

The non-vascularized and uncalcified “cartilage plug” with abnormal growth and differentiation is the most typical pathological manifestation during TD [38]. We first performed histopathological assays to investigate the extent of the osteogenic differentiation of the tibial growth plate. For this purpose, the Safranin-fast green staining (tibia) was used to identify the morphological structure of the osteogenic and non-osteogenic parts (chondrocytes) and the osteogenic tissue. The safranine-fast green staining found obvious morphological differences between the normal and TD group sections on the 10th day. The TD group’s red areas (e.g., non-osteogenic chondrocytes) were significantly larger than those in the normal group, especially the upper and lower span areas at low magnification. While under the high magnification, the width of the growth plate in the TD group extends downward, and the chondrocytes fail to form bone and accumulate downward. The chondrocytes in each area of the tibial growth plate of the normal group had clear boundaries and obvious signs of osteogenesis (Figure 1A). Moreover, we checked the chondrocytes number in hypertrophic and proliferative zones, showing a clear difference between the control and TD groups (Appendix A). To confirm the morphological difference between the growth plate and the chondrocyte partition, the analysis was performed by the Aperio ImageScope (Version: v12.3.2.8013). On the 7th, 10th, and 14th days (*p* < 0.05, *p* < 0.01), it was discovered that the width of the tibial growth plate of the broiler chicks in the TD group was much bigger than that in the normal group. This difference was shown to be statistically significant. The sum of the widths of the quiescent chondrocyte zone and the proliferative zone in the TD group was also significantly larger than that in the normal group (*p* < 0.01) (Figure 1B). Based on the above results, it can be speculated that the pathogenesis of TD in broilers may be due to the inability of chondrocytes in the resting zone and the proliferative zone to transition to the next step and the deposition of numerous cells in the lower area, which eventually leads to the failure of the growth plate to form bone. Following this hypothesis, we conducted follow-up experiments.

### 2.2. The Expression of miR-140-5p Decreased in the Tibial Growth Plate of TD Model

Previous high-throughput sequencing revealed that the expression of miR-140-5p was different in the normal and TD broiler tibial growth plates. The tissue expression specificity of miR-140-5p in various tissues was initially determined using RT-qPCR to investigate the role of miR-140-5p in the growth plate. When compared to broiler heart, liver, spleen, skin, muscle, and other tissues, the expression of miR-140-5p in the growth plate was higher (*p* < 0.01), indicating significant scientific importance (Figure 2A). The broiler TD model was then induced by thiram to validate the sequencing screening findings, the expression profile of miR-140-5p in the tibial growth plate, and the expression changes of miR-140-5p in the normal and TD groups on days 7, 10, and 14 were identified by RT-qPCR. According to the findings, the expression of miR-140-5p in the tibial growth plate of broilers in the TD group fell considerably (*p* < 0.01) on the seventh day as compared to the normal group (Figure 2B). Later, on the 10th and 14th days, the expression of miR-140-5p in the TD group decreased. The expression trend of the animal test findings reflected the earlier sequencing data. Furthermore, the sequencing prediction findings revealed that HDAC4 is a probable target gene of miR-140-5p. The HDAC4 expression mutations were observed in the normal and TD groups at 7, 10, and 14 days to preliminarily confirm the targeting link. The findings revealed that HDAC4 expression in the tibial growth plate rose dramatically on the seventh day of TD; on the fourteenth day, the HDAC4 expression trend was more noticeable (Figure 2C). Furthermore, the HDAC4 protein expression was greater in the TD group than in the control group (*p* < 0.01) (Figure 2D). Consequently, the expression of HDAC4 on the growth plate increased in TD, while miR-140-5p was inversely linked with the translation of HDAC4.

### 2.3. miR-140-5p Promoted the Growth and Differentiation of Growth Plate Chondrocytes

To explore the functional role of miR-140-5p in growth plate chondrocytes, we constructed lentiviral plasmids, LV3-miR-140-5p mimics, and LV3-miR-140-5p inhibitors, and introduced exogenous miR-140-5p into primary growth plate chondrocytes by lentivirus transfection (Figure 3A). The transfected positive chondrocytes were screened by puromycin, and their efficiency was observed by a fluorescence microscope. The RT-qPCR was used to confirm the transfection success by detecting miR-140-5p expression in chondrocytes from each group. The results found that chondrocytes in the mimics group expressed gga-miR-140-5p at a greater level than controls (*p* < 0.01). While compared to the control group and the NC group, there was no substantial change in the expression of gga-miR-140-5p in chondrocytes when using an inhibitor (*p* < 0.01) (Figure 3B). The growth plate chondrocytes were found in different growth and development stages with different morphology. Therefore, the effect of miR-140-5p on the growth and differentiation of chondrocytes was preliminarily observed by cell morphology. The chondrocytes in the control and NC groups were oval, a few were polygonal, and the cells were similar in shape and size and closely arranged (Figure 3C). Therefore, it can be preliminarily judged that the chondrocytes are at this stage of proliferation or transition to the hypertrophic area. The chondrocytes in the mimics group were laid on the bottom of the plate in the shape of a “paving stone”. The cells were uniform in size and closely arranged. At this stage, the cells had typical morphology in the hypertrophic zone. However, the external morphology of chondrocytes in the inhibitor group was roughly oval, with a fuzzy outline and a loose and disordered arrangement, which refers to chondrocytes in a state of growth stagnation with no signs of differentiation. 

The expression of chondrocyte growth and differentiation marker proteins (Col-Ⅱ and Sox9) were analyzed to further clarify the function of miR-140-5p on the growth plate. Compared with the control group, NC group, and mimics group, the expression of Col -Ⅱ and Sox9 protein in the inhibitor group decreased to varying degrees, especially the expression of Sox9 protein (*p* < 0.01). However, the expression was up-regulated in the mimics group. The PTHrP protein is also mainly expressed in chondrocytes in the proliferation area, which primarily promotes the proliferation of chondrocytes. Its expression was substantially reduced in the inhibitor group compared to that in the normal group and the mimics group (*p* < 0.01); however, the expression in the mimics group was significantly higher compared to that in the normal group with no significant difference (*p* > 0.05) (Figure 4A). Then, the regulation of miR-140-5p on chondrocyte differentiation was analyzed by the same means. Runx2 and MEF2C are two crucial transcriptional regulators. They are expressed in large quantities in the transition stage from the proliferative phase to the hypertrophic phase of the growth plate, promoting the hypertrophy of chondrocytes in the growth plate. In the inhibitor group whose expression of miR-140-5p was inhibited, the expression of Runx2 and MEF2C were significantly down-regulated in varying degrees (*p* < 0.01); in the overexpression group of miR-140-5p, the expression of Runx2 and MEF2C showed an upward trend. Col-X, MMP-13, IHH, and COMP proteins are only expressed in hypertrophic chondrocytes, and the expression of Col-X protein indicates that chondrocytes are in the hypertrophic stage. The immunoblotting results showed that Col-X, MMP-13, IHH, and COMP proteins in the chondrocytes of the inhibitor group were in a low expression state. However, the expression of these proteins was up-regulated in chondrocytes of the mimics group with a significant statistical difference (*p* < 0.01). The VEGFA is a key factor in promoting angiogenesis in the growth plate. Its expression means that chondrocytes have differentiated to the terminal hypertrophy stage. Moreover, its expression is decisive in chondrocytes’ final calcification and osteogenesis. Although the expression of VEGFA protein was detected in the chondrocytes of the inhibitor group, the expression was decreased. Compared with the control group and NC group, the expression of the VEGFA protein was significantly up-regulated in the mimics group (*p* < 0.01) (Figure 4B). In conclusion, the presence of the miR-140-5p promoted the expression of chondrocyte growth and differentiation marker proteins. 

The cellular immunofluorescence technique was performed to more directly confirm the expression and cellular localization of the marker proteins. The fluorescence signal of the HDAC4 protein was more robust in the miR-140-5p inhibitor group. The fluorescence signals of proteins such as MEF2C, Runx2, VEGFA, Col-II, and Col-X were almost invisible in the inhibitor group; however, the fluorescence signals of these proteins were evident in the mimics group (Figure 5).

### 2.4. HDAC4 Protein Interacts with Col-Ⅱ, Col-X, and COMP Protein and Hinders the Hypertrophy of Growth Plates’ Chondrocytes

Bioinformatics technology first predicted whether there is a targeting relationship between miR-140-5p and HDAC4. Therefore, the target sequences of miR-140-5p and HDAC4 were searched on the TargetScan 7.2 website. miR-140-5p was found to possess HDAC4 complementary sequences with the same binding sequences for humans and chickens (Appendix A). Next, we searched the miRbase website for the sequence of mature gga-miR-140-5p and the binding sequence targeting HDAC4. Based on the binding sequences, three sequences of HDAC4-140-wild-type, HDAC4-140-mutant and gga-miR-140-5p mimic were designed to construct plasmids. According to the dual-luciferase reporter gene system principle, the 3′UTR sequence of the target gene was assessed with the miRNA. The results showed that the luciferase intensity in the HADC4-WT and miR-140-5p co-transfected group was significantly lower than that in the HADC4-WT and miR-NC as well as the HADC4-MUT and miR-140-5p co-transfected groups (*p* < 0.01) (Figure 6A), confirming that miR-140-5p specifically targets HDAC4. 

We have demonstrated that miR-140-5p targets and inhibits the translation of HDAC4 mRNA and further affects the expression of HDAC4-interacting proteins to regulate chondrocytes’ function. To explore the mechanism of action, we performed co-immunoprecipitation to verify the existence of proteins interacting with HDAC4 and screened out the proteins interacting with HDAC4. Before CO-IP, we detected the target signal by Western blot. The target protein and the reference protein bands were detected, but there were a few hetero bands. Considering that the antibodies used in this experiment were all rabbit polyclonal antibodies, the small number of hetero bands does not affect this study. The CO-IP results showed that the bait protein HDAC4 was detected in the IP group, having numerous differential proteins compared with the IgG group, which could be detected by mass spectrometry (Figure 6B). The above proteins were analyzed and evaluated by LC-MS/MS, and the obtained data were directly submitted to the ProteinPilot (version: v5.0) for database retrieval. The analysis after mass spectrometry identified the specific proteins. When the reliability conf ≥ 95% and unique peptides ≥ 1, the total number of proteins identified in the HDAC4_IP and HDAC4_IgG protein samples were 470 and 5, respectively (Appendix A). The functional annotation and enrichment pathway analysis of high-confidence proteins showed that Col-X, Col-II, and COMP proteins interacting with HDAC4 proteins participated in the differentiation and development of chondrocytes. 

To determine whether Col-X, Col-II, and COMP proteins interact with HDAC4, the study used HDAC4 specific inhibitor LMK-235 and agonist ITSA-1 to treat growth plate chondrocytes and detect HDAC4 interacting proteins’ expression of chondrocytes growth and differentiation. To ensure that the viability of chondrocytes has been in a good state during the treatment of HDAC4 inhibitors and agonists, the optimal concentration range of reagents was first evaluated by cytotoxicity test. After the chondrocytes were treated with the medium containing different concentrations of ITSA-1 for 24 h, the chondrocyte viability showed a downward trend with the increase in concentration; when the maximum concentration of 200 nM was used, the chondrocyte viability was the lowest, but it was also above 60%. The effect of LMK-235 on chondrocyte viability was like that of ITSA-1. When the maximum concentration of 150 nM was used, the chondrocyte viability was about 60%. The recommended concentration of the reagent can ensure that the viability of chondrocytes is above 60%, which meets the requirements of the subsequent test (Figure 6C); however, to reduce the interference of cytotoxicity on the results, the final concentration of ITSA at 50 nM, 100 nM, and 150 nM and LMK-235 at 25 nM, 50 nM, 100 nM were selected in the study.

After chondrocytes were treated with 50 nM LMK-235 and 100 nM ITSA-1 for 24 h, an inverted microscope first observed the morphological changes in the chondrocytes in the two groups. The chondrocytes in the LMK-235 group were polygonal in the shape of “paving stones”, the outlines of the cells were three-dimensional and closely arranged, and the cells had a grainy feel inside. However, chondrocytes in the ITSA-1 group were oblate or oval, with different cell sizes and a loose and disordered arrangement. Preliminary studies on the morphological structure of chondrocytes in different stages can preliminarily speculate that chondrocytes in the LMK-235 group were in the transitional or hypertrophic stage, while chondrocytes in the ITSA-1 group were still in the proliferative stage (Figure 6D). 

To further explain that the two groups of chondrocytes have different external shapes, this experiment revealed the potential mechanism by detecting the expression of the two reagents on the chondrocyte growth and differentiation marker proteins. The results of Western blotting and grey value analysis showed that the protein expression of HDAC4 in chondrocytes of the LMK-235 group was significantly decreased, but there was no correlation between the expression of protein and the concentration of reagents. The expression of HDAC4 in the chondrocytes of the ITSA-1 group was significantly increased. Protein expression was reduced, but again was not significantly correlated with reagent concentration. First, the protein expression levels of Col-II, Col-X, and COMP proteins interacting with HDAC4 in the LMK-235 group were significantly higher than those in the 0 nM negative control group (*p* < 0.01). The protein expression levels were decreased to different degrees compared to the control group, while some concentrations were significantly different from the control group (*p* < 0.01). The protein expression of Sox9, which is expressed in the growth plate proliferation zone, was significantly higher in the chondrocytes of the LMK-235 group than in the ITSA-1 group (*p* < 0.01). To varying degrees, the protein expressions of Runx2 and MEF2C in the ITSA-1 group were lower than those in the LMK-235 group (*p* < 0.01). Finally, the protein expression levels of MMP13 and VEGFA expressed in the growth plate’s hypertrophic and terminal hypertrophic regions were at lower levels in the ITSA-1 group. Hence, the expression of the HDAC4 protein and the marker proteins are associated with the growth and differentiation of chondrocytes (Figure 7A).

We further verified the marker proteins’ expression trend and protein expression localization by cellular immunofluorescence. As shown in Figure, the fluorescence signal of HDAC4 in the chondrocytes of the LMK-235 group was extremely weak, while the fluorescence of Col-X, MEF2C, COMP, and Col-II is remarkably high (Figure 7B). The results of this experiment were consistent with Western blotting. In conclusion, HDAC4 protein interacts with Col-X, COMP, and Col-II proteins and inhibits their expression.

Whether the above results are also valid on the TD animals’ model, the TD was induced in chickens using thiram. The normal and TD groups were detected on the 10th day when the broiler chickens showed typical clinical symptoms. The results showed that the protein expressions of Runx2 and MEF2C were decreased to varying degrees in the TD group, while the protein expressions of Col-II, Col-X, and COMP, and HDAC4 showed the opposite trend with a more significant trend of Col-X and COMP proteins in the TD group (*p* < 0.01) (Figure 8).

### 2.5. HDAC4 Expression Decreased during Differentiation of Growth Plate Chondrocyte Dynamics

The above studies revealed the potential mechanism of HDAC4 on the growth and differentiation of chondrocytes by inhibiting and activating the activity of HDAC4. However, the change in HDAC4 protein expression activity during the normal differentiation of chondrocytes was unknown. Therefore, we monitored the dynamic expression of related marker proteins in the first, third, fifth, and seventh days of the chondrocyte growth cycle. Western blotting results showed that the expression of HDAC4 protein in chondrocytes in ITS medium showed a dynamic decline, and the protein expression on the fifth and seventh days of culture was significantly lower than the initial state (*p* < 0.01). However, intramembrane osteogenesis-related target protein BMP2 expression was up-regulated on the third and fifth days in the ITS medium compared with the control group and significantly decreased on the seventh day. Coincidentally, this expression trend was like the protein expression trend of the pro-chondrocyte differentiation markers Runx2 and MEF2C, combined with the growth characteristics of early chondrocytes in the proliferative phase and transition phase (pre-hypertrophy) from day 3 to day 5 (Figure 9A). Therefore, these results reveal the dynamic expression of related target proteins and reflect the expression function of these proteins, which is in line with the essential characteristics of chondrocyte differentiation. Finally, we studied the markers of chondrocyte hypertrophy and terminal calcification. The results showed that Col-X, COMP, IHH, and VEGFA proteins in chondrocytes in ITS medium were all at a low expression or no expression before the third day state; and after that, it was in a state of high expression (Figure 9B). In conclusion, the up-regulated expressions of Runx2, MEF2C, Col-X, COMP, and other proteins promoted and dominated the differentiation of chondrocytes. However, the decrease in HDAC4 protein expression reversely proved that its existence was not conducive to the transition and differentiation of chondrocytes.

### 2.6. Total Flavonoids of Rhizoma Drynariae Promoted the Differentiation of Growth Plate Chondrocytes via Up-Regulating the Expression of miR-140-5p

The total flavonoids of Rhizoma drynariae (TFRD) is a drug which has been traditionally used in China to treat musculoskeletal disorders [39]. Our previous research found that TFRD effectively restored the loss of the tibial growth plate. We wondered whether the total flavonoids of *Rhizoma drynariae* up-regulated or inhibited the expression of miR-140-5p in growth plates. The expression of miR-140-5p and HDAC4 in chicken tibial growth plates during the 10th and 14th days of the TD and TFRD groups were compared to evaluate the HDAC4 expression. Similarly, to the previous experiment, we found that miR-140-5p expression in the tibial growth plate of the TD group was significantly lower than that of the control group. In the group with TFRD, its expression was significantly lower than in the comparison group. TFRD significantly up-regulated the miR-140 expression compared to the TD group (*p* < 0.01). However, miR-140-5p showed a negative trend in the tibial growth plate of HDAC4 mRNA expression. A substantial difference was seen on day 14 in HDAC4 mRNA expression levels in the tibial growth plates of chickens from the TD and TFRD groups, even though TFRD expression levels were also more significant than those in the normal group. The TFRD group chickens recovered well after treating the total flavonoids of Rhizoma drynariae. At the same time, the expression of miR-140-5p was up-regulated and the expression of HDAC4 was down-regulated in the growth plate (Figure 10A). Studies have found that the total flavonoids of *Rhizoma drynariae* can directly act on osteoblasts and osteoclasts to promote bone metabolism and recovery from bone injury. We further speculated whether the total flavonoids of *Rhizoma drynariae* promoted the growth and differentiation of chondrocytes in the growth plate, thereby realizing the process of endochondral osteogenesis in the growth plate. The expression of growth-promoting chondrocyte growth markers and chondrocyte differentiation-promoting markers were detected on the 10th and 14th days. In TD, the HDAC4 protein was highly expressed in tibial growth plate chondrocytes, and the expression level was significantly higher than that in the normal and treatment groups; however, the expression levels of BMP2/Runx2, the key factor in the intramembrane osteogenic signal pathway, and IHH/PTHrP, the target factor of endochondral osteogenesis, were low, and even the hypertrophy marker proteins COMP and Col-X of growth plate chondrocytes were almost not expressed. It is obvious that TD seriously affected the transition and differentiation of chondrocytes. After administration with total flavonoids of *Rhizoma Drynariae*, the obvious finding was that the expressions of BMP2, Runx2, MMP13, Col-X, and COMP were up-regulated, but the expression levels did not reach the normal level. In addition to HDAC4, the protein and gene expression levels of other markers related to chondrocyte growth and differentiation in the treatment group were significantly higher than in the TD group (*p* < 0.01 or *p* < 0.001). In particular, the expression of chondrocyte hypertrophy markers Col-X, matrix metalloproteinase MMP13, cartilage matrix protein COMP, transcription factors MEF2C, and Runx2 promoted chondrocyte differentiation in the TFRD group, which was significantly higher than the TD group and normal group (Figure 10B–D).

Briefly, the total flavonoids of *Rhizoma Drynariae* promoted the growth and differentiation of chondrocytes by regulating the signal factors of intramembrane osteogenesis and endochondral osteogenesis.

## 3. Discussion

Skeletal development is accomplished through two modes, i.e., intramembranous osteogenesis and endochondral osteogenesis. Endochondral osteogenesis constitutes most of the body’s long bones and determines the length of the bone and body [40,41]. Chondrocytes originate from mesenchymal stem cells and undergo proliferation, hypertrophy, calcification, and ossification. Therefore, the normal growth and differentiation of chondrocytes are crucial for the chondrogenesis of the tibial growth plate [42]. Due to their high degree of conservation, microRNAs participate in the key regulation of various functions such as cell differentiation, proliferation, and apoptosis [2]. We previously reported that TD resulted in an extraordinary expression of numerous miRNAs in the growth plate, especially miR-140-5p [27]. In this study, we demonstrated that the targeted inhibition of HDAC4 expression by miR-140-5p coordinated the hypertrophic differentiation of growth plate chondrocytes and restored the process of endochondral osteogenesis. 

Numerous studies have shown that miR-140 is mostly found in various tissues but is highly expressed in mammalian chondrocytes. It regulates chondrocyte differentiation, growth, and homeostasis [29,43]. Giedre and coworkers examined the miR-140 gene; they found a unique mutation in the seed region that causes an autosomal dominant human chondrocyte dysplasia with a distinct morphological appearance. [30]. Our findings support earlier research findings that miR-140-5p is abundant and specific in the chondrocytes of growth plates. It was further speculated that the decrease in miR-140-5p expression caused the increase in HDAC4 expression, which disrupted the homeostasis of growth and differentiation of growth plate chondrocytes. At present, the mainstream methods of cell transfection include electroporation, liposome transfection, virus-mediated transfection, etc. In the early stage, the liposome transient transfection method was tried, and the success rate of transfection could reach 70%. However, due to the high toxicity of transfection reagent to chondrocytes, the state of cells was poor, and even some cells were lysed. Therefore, the lentiviral transfection method was finally adopted in the current study. Although the transfection efficiency is general, the positive cells can satisfy the subsequent experiments after drug screening. Extensive references have also shown that primary cells reject heterologous substances due to their inherent conservation and stability, resulting in difficulties in the transfection of primary cells and low transfection efficiency [44,45]. Growth plate chondrocytes at different stages not only have great differences in morphology but also differ in driving function, having their expressive markers [40]. Col- II, Sox9, and PTHrP are expressed in proliferative chondrocytes to maintain the chondrocyte phenotype and promote cell growth [46]. Runx2 and MEF2C are key transcription factors that promote chondrocyte differentiation, and their expression directly affects the development of cartilage, while Col-X, MMP-13, and IHH are markers of hypertrophic chondrocytes, dominating the final chondrogenesis of chondrocytes, especially VEGFA [47]. This study found that the expression of miR-140-5p in chondrocytes of the growth plate was silenced, and the expression of the chondrocyte growth and differentiation marker protein was also down-regulated, but HDAC4 was at a high expression level. Previous studies only reported that miR-140-5p was closely related to cartilage formation but did not clarify the specific function. This study confirmed that the suppression of miR-140-5p would lead to the failure of osteogenesis in the growth plate but also revealed that miR-140-5p could promote the growth and differentiation of growth plates’ chondrocytes.

The miRNAs play a biological regulatory role in animal development by binding and pairing with some bases of target mRNA, leading to silence or blocking the translation process of target mRNA [48,49]. Current research reports that miR-140-5p has a direct targeting relationship with Wnt, Sox, and VEGF and acts on different disease models. We first confirmed that miR-140-5p directly and specifically targeted HDAC4 through the double luciferase reporter gene system, and then we confirmed its involvement in the associated proteins and gene expression. Histone deacetylases (HDACs) play a non-negligible role in regulating gene expression by changing the structure of chromosomes in cells [31]. It belongs to class II HDACs and has been closely related to the hypertrophy of chondrocytes, affecting the growth and differentiation of growth plate chondrocytes [32,33]. According to studies, HDAC4 interacts with Runx2 to limit the production of Runx2 protein, delaying the hypertrophic differentiation of chondrocytes and the development of endochondral osteogenesis [36]. Additionally, it has been shown that MEF2C enhances the Runx2 expression and chondrocyte hypertrophy by acting upstream of Runx2 [50]. However, HDAC4 inhibits chondrocyte hypertrophy mediated by the MEF2C transcription factor. We found that the HDAC4 protein interacted with Col- II, Col-X, and COMP protein in broiler tibial growth plate chondrocytes, while no interaction was found with MEF2C, Runx2, and HDAC4. However, by inhibiting the expression of HDAC4, we confirmed that the expression of Col-II, col-X, and COMP proteins was up-regulated in varying degrees, and the expression of MEF2C and Runx2 was also significantly up-regulated. At the same time, Western blot results showed that HDAC4 inhibited the expression of MEF2C, Runx2, Col- II, Col-X, and COMP proteins in broiler chondrocytes, thus hindering the hypertrophy of chondrocytes, which is consistent with the previous research conclusions. 

The total flavonoids of Rhizoma drynariae (TFRD) are the monomers of traditional Chinese medicine extracted from the traditional Chinese medicine Rhizoma drynariae [39]. Researchers have found that total flavonoids of Rhizoma drynariae play an important role in bone repair and remodeling by promoting osteoblasts’ osteogenic capacity and inhibiting osteoclast clearance [51]. Our previous studies also confirmed that TFRD targeted the BMP2/Runx2 and IHH/PTHrP signaling axes to achieve promising therapeutic effects on TD [4,52]. This study demonstrated that the total flavonoids of *Rhizomadrynariae* promoted the growth and differentiation of growth plate chondrocytes and the formation and invasion of growth plate angiogenesis via up-regulating the expression of miR-140-5p and inhibiting the expression of HDAC4 (Figure 11). At present, researchers have administrated TFRD in different skeletal diseases in sheep, rabbits and rats and achieved good therapeutic effects, targeting the Smad/BMP signaling pathway to promote the proliferation and differentiation of osteoblasts [53,54].

The current study has several limitations including the animal models of the natural onset of TD, which are challenging to sample, and the age of onset varies, which interferes with the results. However, we previously proved that the clinical manifestations, pathological changes, and differential gene expression of thiram-induced TD and naturally occurring TD are remarkably similar. This study has screened and verified the interacting proteins of HDAC4, the direct or indirect interaction of HDAC4 with other associated proteins, and the potential mechanism by which HDAC4 affects the expression of interacting proteins, which would need to be further defined.

## 4. Material and Methods

All methods were approved and conducted by the Hubei Laboratory Animals Research Centre in the People’s Republic of China and the Huazhong Agricultural University’s Ethics Committee (Permit number: 4200695757). All animal studies and methods were conducted in accordance with the relevant requirements of China’s Proclamation of the Standing Committee of the Hubei People’s Congress (PSCH No. 5). 

### 4.1. Antibodies

The following antibodies were used: rabbit anti-COL2A1 (1:1500 for WB, 1:200 for IHC/IF, ABclonal, A1560), rabbit anti-Aggrecan (1:1500 for WB, 1:200 for IHC/IF, ABclonal, A8536), rabbit anti-IHH (1:1500 for WB, ABclonal, A6626), rabbit anti-SOX9 (1:1500 for WB, 1:200 for IHC/IF, ABclonal, A2479), rabbit anti-PTHLH (1:1500 for WB, 1:200 for IHC/IF, ABclonal, A12492), rabbit anti-HDAC4 (1:1500 for WB, 1:200 for IHC/IF, ABclonal, A0179), rabbit anti-HDAC4 (1:100 for IP, ABclonal, A0239), rabbit anti-CMP (1:1500 for WB, 1:200 for IHC/IF, ABclonal, A13963), rabbit anti-VEGF (1:1500 for WB, 1:200 for IHC/IF, ABclonal, A12303), rabbit anti-RUNX2 (1:1000 for WB, 1:200 for IHC/IF, ABclonal, A2851), rabbit anti-MEF2C (1:1500 for WB, 1:100 for IHC/IF, ABclonal, A12385), rabbit anti-MMP13 (1:1500 for WB, 1:200 for IHC/IF, ABclonal, A11148), rabbit anti-Collagen X/COL10A1 (1:1500 for WB, 1:200 for IHC/IF, ABclonal, A18604), rabbit anti-BMP2 (1:1500 for WB, 1:200 for IHC/IF, ABclonal, A0231), rabbit anti-GAPDH (1:5000 for WB, 1:00 for IHC/IF, 1:50 for IP, ABclonal, AC001), rabbit anti-β-Actin (1:10,000 for WB, ABclonal, AC026), HRP Goat Anti-Rabbit IgG (H + L) (1:5000 for WB, ABclonal, AS014), Fluorescein(FITC)-conjugated Affinipure Goat Anti- Rabbit IgG (H + L) (1:100 for IF, proteintech, SA00003-2), Cy3-conjugated Affinipure Goat Anti- Rabbit IgG (H + L) (1:100 for IF, proteintech, SA00009-2).

### 4.2. Animal Model Experiment

A total of 180 one-day-old arbor acres (AA) chickens were separated into three groups: cell culture, control, and TD. The animals were randomly assigned to separate experimental groups to avoid bias and ensure an even distribution of possible confounding variables. Additional controls were implemented by carefully choosing animals from the same breed, age, and sex for each experimental group. The cell culture and control groups were given a usual diet with unrestricted access to water, while the TD group was provided a full-fledged meal with 50 mg/kg thiram from the third to the fourteenth day (Macklin, Shanghai, China). The feeding cycle lasted 14 days (Appendix A). On days 7, 10, and 14, and 20 chickens from the control and TD groups were randomly chosen for cell culture. After three days of normal feeding, 240 one-day-old arbor acres (AA) chickens were randomly separated into three further groups (e.g., control group, TD group, and TFRD group) for the in vivo model. The chickens in the TD group were given 50 mg/kg thiram from the third to the fourteenth day to induce TD. From the third through the seventh day of the trial, the chickens in the TFRD group received 50 mg/kg of thiram in their meal. They were then switched to their usual feed and given 20 mg/kg/day of total flavonoids from Rhizoma drynariae; control group broilers were provided with their regular diet during the period (Appendix A). The daily lighting time was fixed at 23 h and 1 h of lights-off time. During the experiment, the temperature and humidity of the chicken house, as well as the health of the flock, were strictly controlled. Furthermore, half of the tibia samples were placed in 4% paraformaldehyde (Solar bio, Beijing, China) for hematoxylin and eosin staining, safranin-fixed green staining, and immunofluorescence staining, while the other half were stored in liquid nitrogen at −80 °C for RT-qPCR and Western blot analysis.

### 4.3. Cell Line and Growth Plate Chondrocytes’ Primary Culture

The 239T cell line was obtained from the Chinese Academy of Sciences Cell Bank and cultured in DMEM-F12 media (Gibco, Billings, MT, USA) supplemented with 10% Australian fetal bovine serum (Gibco, Billings, MT, USA). After the eighth day, growth plate chondrocytes were extracted from the proximal tibial growth plate of AA chickens. Under sterile circumstances, the growth plate tissues were dissected from the tibia. After 11.5 h of digestion in 0.1 percent collagenase type IV (Biotop, Shanghai, China) and 0.1 percent hyaluronidase (Biotop, Shanghai, China), chondrocytes were filtered through a 200-mesh cell sieve (Corning, NY 14831, USA), seeded in 2 × 10^5^ cells/mL per well, and grown at 37°C with 5% CO_2_ [4]. The chondrocyte culture media was made up of 87 percent high-glucose DMEM medium (Gibco, Billings, MT, USA), 12 percent Australian fetal bovine serum (Gibco, Billings, MT, USA), and 1 percent penicillin–streptomycin (Gibco, Billings, MT, USA). Furthermore, a regular culture medium and 1 ITS supplement was put in the ITS medium (Cyagen, Wuhan, China).

### 4.4. Construction of Retroviral Vector and Lentivirus Transfection

#### 4.4.1. Lentivirus Packaging

The 293T cell suspension was cultured in a 15 cm culture dish with 10% FBS-containing DMEM medium at 37 °C with 5% CO_2_ overnight. The 1.5 mL serum-free DMEM was added into a 5 mL sterile centrifuge tube, and shuttle plasmids (GenePharma, Shanghai, China) containing gga-miR-140-5p mimics, gga-miR-140-5p inhibitors, and LV3-NC sequences and packaging plasmids (pGag/Pol, pRev, pVSV-G) (GenePharma, Shanghai, China) were added in proportion. Afterward, 1.5 mL serum-free DMEM was mixed with 300 μL RNAi-Mate (GenePharma, Shanghai, China) in another sterile 5 mL centrifuge tube. The above two tubes were mixed and placed for 25 min. The cell supernatant in the culture dish was aspirated into a 50 mL centrifuge tube at 4000 rpm at 4 °C for 4min. After the ultracentrifugation of the filtrate with 20,000 rpm at 4 for 2h, the concentrate was collected, aliquoted, and stored at −80 °C. 

#### 4.4.2. Lentivirus Titer Detection

The 10 μL lentivirus stock solution was diluted to 10-fold with 10% FBS DMEM medium for 3–5 gradients. The fluorescent cells were counted in the 96-well plate by fluorescence microscope or FACS to calculate the virus titer in combination with the dilution factor (the final virus titer was not less than 8 × 10^8^ TU/mL). 

#### 4.4.3. Lentivirus Transfection

The second-generation growth plate chondrocytes in good condition were digested, resuspended, and then seeded into a 12-well plate at 37 °C with 5% CO_2_ overnight. Then, gga-miR-140-5p mimics, gga-miR-140-5p inhibitors, and LV3-NC were mixed and diluted 1:100 with the culture medium. The total volume was about 1 mL with the final concentration of 5 μg/mL Polybrene. The medium containing the virus liquid was added to the 12-well plate. After 24 h, the virus dilution liquid was sucked and replaced with a normal medium at 37 °C with 5% CO_2_ for 72 h.

### 4.5. Safranin-Fixed Green Staining and Hematoxylin and Eosin (H&E) Staining

Tibia cartilage in each group was decalcified in a neutral EDTA-2 Na solution (Solar bio, Beijing, China). After 20 days, the decalcification was terminated, dehydrated, and transparent with high-concentration ethanol and xylene and embedded with paraffin. The paraffin sections were successively deplastized by ethylene glycol, ethyl ether acetate, and high-concentration ethanol. The sections were put into the fast green dye solution for 3–4 min. The excess dye solution was washed away with water. Then, sections were placed into a safranin staining solution for 15–30 s. Finally, xylene was rinsed for 5 min and then the slide was mounted with neutral gum. The sections were stained with hematoxylin staining solution for 8–10 min, rinsed with tap water, dehydrated in 85 percent and 95 percent gradient alcohol for 5 min each, and then stained for 8–10 min with eosin staining solution. The slices were passed through 100% ethanol, xylene, and neutral gum before being sealed.

### 4.6. Immunofluorescence Staining and Immunohistochemistry

The cells were fixed with 4% paraformaldehyde for 15 min after immersing the adherent cells with PBS (HyClone, South Logan, UT, USA). Then, 0.5% TritonX-100 (biosharp, Hefei, China) diluted with PBS was used to penetrate the cell membrane for 20 min. A 5% albumin bovine V (Servicebio, Wuhan, China) was added to the block for 30 min. After pouring out the blocking solution, the diluted primary antibody working solution was directly added and incubated at 4 °C overnight. After that, the residual primary antibodies were washed away with PBS, and the diluted fluorescent secondary antibody was added at room temperature for 1 h. Finally, the DAPI nuclear staining solution (biosharp, Hefei, China) was added for 10 min in the dark. The proteinase K antigen retrieval was performed on paraffin sections for tibial growth plate tissue, while the other steps were the same as the above. 

### 4.7. Dual-Luciferase Reporter Assay

The 293T cells were seeded into a 12-well plate at the concentration of 5 × 10^5^ cells/well at 37 °C with 5% CO_2_ before transfection. Then, cells were co-transfected with 4 μL of miR-140-5p mimics, miR-NC (GenePharma, Shanghai, China) and 50μL Opti-MEM Ⅰ (Gibco, Billings, MT, USA), together with 2 μg pGLO-HDAC4-WT, pGLO-HDAC4-MUT, pGLO-HDAC4-NC (GenePharma, Shanghai, China), and 50μL Opti-MEM Ⅰ with 2μL/well lipofectamine 3000 transfection reagent (Life Technologies, Carlsbad, CA, USA). Five hours after transfection, a complete medium containing FBS was replaced for the next 24 h. Lastly, luciferase activity was determined using a dual-luciferase assay kit (Promega, Madison, WI, USA) following the manufacturer’s instructions. Firefly and Renilla luciferase activity was observed on the microplate reader. After subtracting the background value (blank) from the fluorescence value of each group, we compared the fluorescence value of Firefly to the fluorescence value of Renilla and then NC as the reference for statistics.

### 4.8. Agonist and Inhibitor Assay

The agonist of HDAC4 is ITSA-1 (GlpBio, Montclair, CA, USA), and the inhibitor of HDAC4 is LMK-235 (GlpBio, Montclair, CA, USA, GC13237). Growth plate chondrocytes were seeded into a 6-well plate at 37 °C with 5% CO_2_ one day before administration. The chondrocytes were treated with ITSA-1 and LMK-235 with concentrations of 0 nM, 25 nM, 50 nM, 100 nM and 150nM for 24 h [55,56]. Finally, after observing external morphology with an inverted microscope, the total RNA and protein were extracted with some other cells’ samples, and fixed with 4% paraformaldehyde.

### 4.9. Co-Immunoprecipitation and Mass Spectrometry 

At 4 °C, the lysate was applied to the cell culture plates for complete cell lysis. The lysates were centrifuged at 12,000 rpm for 10 min to recover the supernatant. A small fraction of lysate was collected and stored for future use. The leftover lysate was treated overnight at 4 °C with the HDAC4 antibody (10 g). PierceTM Protein A/G Agarose Beads (Thermo Fisher Scientific, Waltham, MA, USA) were used several times to rinse the lysis buffer. The pretreated beads were added to the cell lysate, incubated overnight at 4 °C, and then centrifuged for 3 min at 2500 rpm at 4 °C. The samples were then incubated at 95 °C for 5 min before being subject to WB analysis in 100 µL of 2 × SDS loading buffer. For mass spectrometry, trypsin was used to contact the rubber particles after cleaning and dehydrating the cut rubber particles (colloidal particle diameter 1–2 mm) with 25 mM NH4HCO3. The Ultraflex III mass spectrometer (Bruker, Mannheim, Germany) was used to conduct analysis for the following parameters: reflection mode, ion source acceleration voltage 1 was 24 kv; acceleration voltage 2 was 22 kv; ion delay extraction was 0.000 ns; vacuum degree was 1.4 × 10^−7^ Torr; mass spectrum signal single scanned accumulation 200 second-rate. The standard Maker peak was used as an external standard to calibrate the mass spectrum peaks, determine the positive ion spectrum, and control the measurement range within 700–4000. The LIFT technique (MALDI LIFT-TOF/TOF MS) performed cascade mass spectrometry on the 5 peaks with the highest PMF intensity (peak intensity greater than 300).

### 4.10. Immunoblotting

RIPA lysate (Solar bio, Beijing, China) was used to recover total proteins from the cell culture plate. The extracted protein concentrations were measured through an OD562 nm microplate reader and the Bicinchoninic acid assay kit (Beyotime, Jiangsu, China). The Bio-Rid protein transfer device was used to transfer the protein to a PVDF membrane (Immobilon, Stockbridge, GA 30281, USA). The primary antibody (rabbit polyclonal antibody) was incubated overnight at 4 °C on the protein-membrane after blocking it for one hour at 25 °C with 5% skim milk in TBST buffer (BD Biosciences, San Jose, CA, USA). Then, the goat anti-rabbit secondary antibody was added and shaken for 1 h at room temperature. Finally, the Fusion software (Viber, Beijing, China) was used to examine the bands of all proteins at the end.

### 4.11. RNA Extraction, Reverse Transcription, PCR, and RT–qPCR

After obtaining total RNA from growth plate chondrocytes with the help of Trizol (Invitrogen, Waltham, MA, USA), the cDNA was transcribed using a cDNA synthesis kit (TaKaRa, Shiga, Japan). As a template for gene sequencing, primary cell cDNA was used, and 2 × Taq Master mix as well as the required primers (Appendix A). Electrophoresis on a 3% agarose gel was used to identify the 5 μL of PCR product. Using the Step One-PlusTM RT-qPCR system, the RT-qPCR was performed in quadruplex with reaction conditions, e.g., denaturation at 94 °C for 30 s and 5 s, annealing at 58 °C for 15 s, and extension at 72 °C for 30 s for 40 cycles of amplification (Applied Biosystems, Woburn, MA, USA). The growth plate was ground into a powder with liquid nitrogen in an enzyme-free mortar for tibial growth plate tissue. Then, the powder was added to Trizol to extract the total RNA by following the steps as mentioned above. For miRNA, before reverse transcription, we designed and synthesized miRNA primers with stem-loop structures. Then, these were transcribed into cDNA with the miRNA exclusive cDNA synthesis kit (GenePharma, Shanghai, China). In addition, the internal reference quantification of mRNA was conducted using GAPDH, while the internal reference quantification of miRNA was conducted using U6. The 2^−ΔΔCt^ technique was used for gene expression analysis.

### 4.12. Statistical Analysis

At least three separate trials were conducted for each experiment. The data were represented with the help of the mean and standard deviation. Using the GraphPad Prism 7 program (version; 8.0.2), we analyzed the differences between the groups by performing a Student t-test, a one-way variance analysis, and Dunnett’s multiple comparison test. In this study, *p* values of 0.05 or below were considered to indicate statistical significance.

## 5. Conclusions

The differentially expressed miR-140-5p in the tibial growth plate is closely related to the growth and differentiation of chondrocytes. Our findings showed that miR-140-5p coordinated the growth and differentiation of growth plate chondrocytes by targeting HDAC4. Furthermore, TFRD effectively restored TD by up-regulating miR-140-5p and inhibiting HDAC4 to improve the differentiation and development of growth plate chondrocytes.

## Figures and Tables

**Figure 1 ijms-24-10975-f001:**
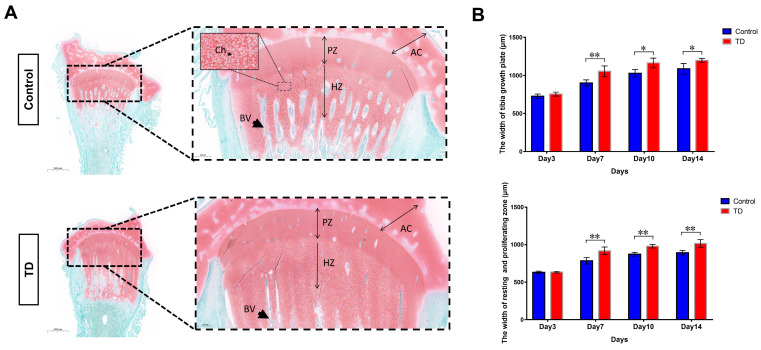
Histopathological analyses of tibia in chickens. (**A**): Safranin-fast green staining of tibia paraffin sections showed increased sulfated glycosaminoglycan deposition in the thiram group. The areas in the dotted line are the resting and proliferative zones of the growth plate, and the black arrow points to the extended proliferation zone (scale bars, 2 mm left and 500 μm right). AC; articular cartilage, PZ; proliferative zone, HZ; hypertrophic zone, BV; blood vessel, Ch; Chondrocyte. (**B**): The width of the tibial growth plate in the resting and proliferative zones in the growth plate was measured by the pathological section analysis software, i.e., Alperio ImageScope (Version: v12.3.2.8013). Five slices were measured in each group, and each slice was measured three times. (* *p* < 0.05, ** *p* < 0.01).

**Figure 2 ijms-24-10975-f002:**
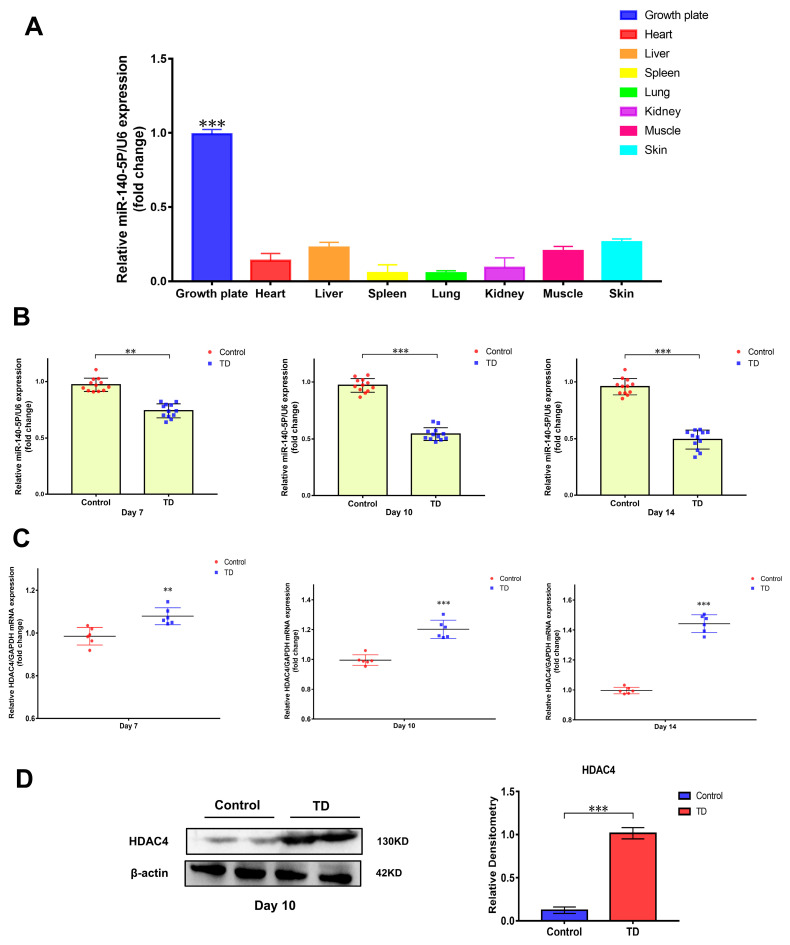
Verification of the difference of miR-140-5p expression in the pathological model and tissue expression specificity. (**A**): Determination of the tissue expression profile of miR-140-5p, using U6 as the housekeeping gene to detect the relative expression of miR-140-5p in different tissues by RT-qPCR. (**B**): Validation of levels of miR-140-5p in growth plates of control and TD groups on the 7th, 10th, and 14th days. (**C**): Validation of levels of HDAC4 mRNA in growth plates of the control and TD groups on the 7th, 10th, and 14th days, using GAPDH as the housekeeping gene. (**D**): Validation of levels of HDAC4 protein in the growth plates of the control and TD groups on the 10th day, using β-actin as an internal reference protein. Repeated five times for each group (** *p* < 0.01, *** *p* < 0.001).

**Figure 3 ijms-24-10975-f003:**
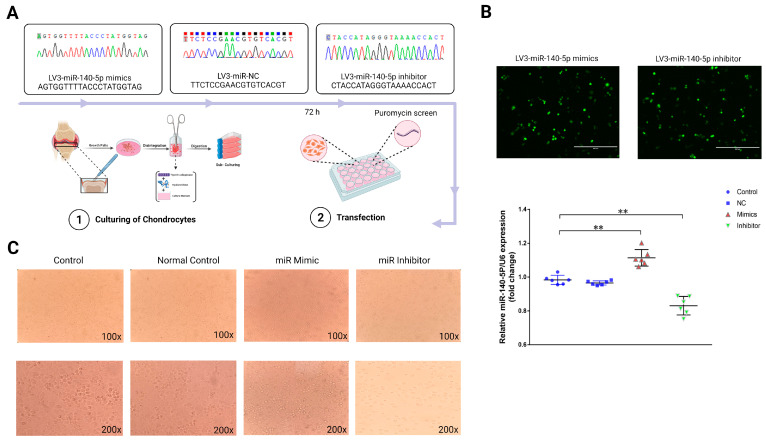
Functional study of miR-140-5p on growth plate chondrocytes. (**A**): Schematic diagrams of lentiviral transfection of chondrocytes. Shuttle and packaging plasmids (pGag/Pol, pRev, pVSV-G), Screening with puromycin after 72 h of transfection. (**B**): Validation of the lentiviral transfection efficiency. Cells with green fluorescence were successfully transfected chondrocytes (Scale bars, 400 μm). The expression of miR-140-5p in each group was determined by RT-qPCR, using U6 as the housekeeping gene. (**C**): The external morphological analysis of chondrocytes in each group after 72 h of transfection (upper is 100X, down is 200X). “*” shows significance level between groups ** *p* < 0.01.

**Figure 4 ijms-24-10975-f004:**
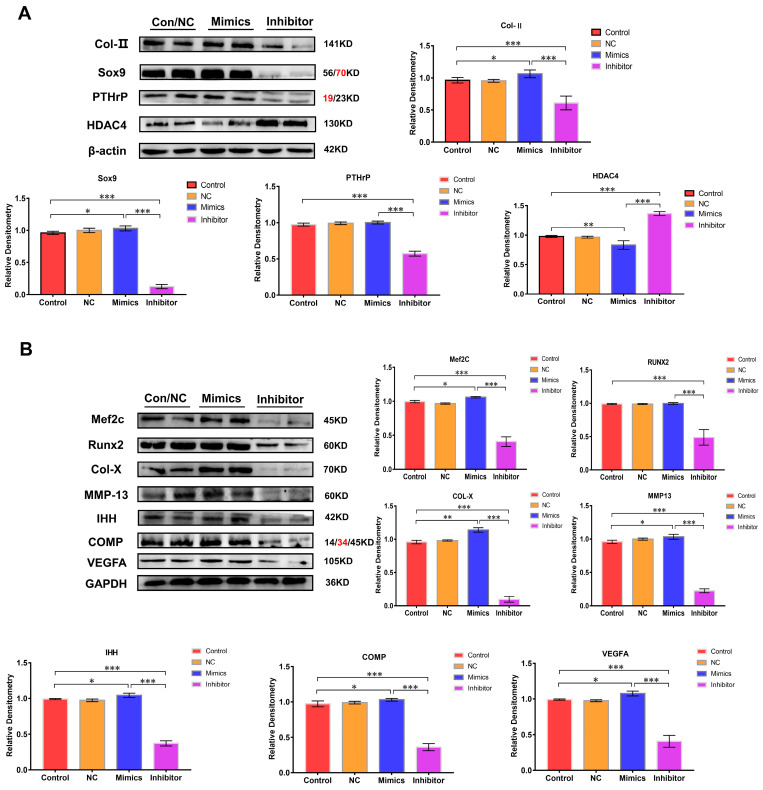
The proteins expression level through Western blotting. (**A**): Analysis of expression levels of marker proteins (Col-II, Sox9, PTHrP, and HDAC4) in chondrocytes at different stages (growth and differentiation), using β-actin and GAPDH as internal reference proteins. (**B**): Analysis of expression levels of marker proteins (MEF2C, Runx2, Col-X, MMP-13, IHH, COMP, and VEGFA) in chondrocytes at different stages (growth and differentiation), using β-actin and GAPDH as internal reference proteins. The red text refers to a distinct band corresponding to targeted molecular weight (MW). “*” shows the level of significance among groups. * *p* < 0.05, ** *p* < 0.01, *** *p* < 0.001.

**Figure 5 ijms-24-10975-f005:**
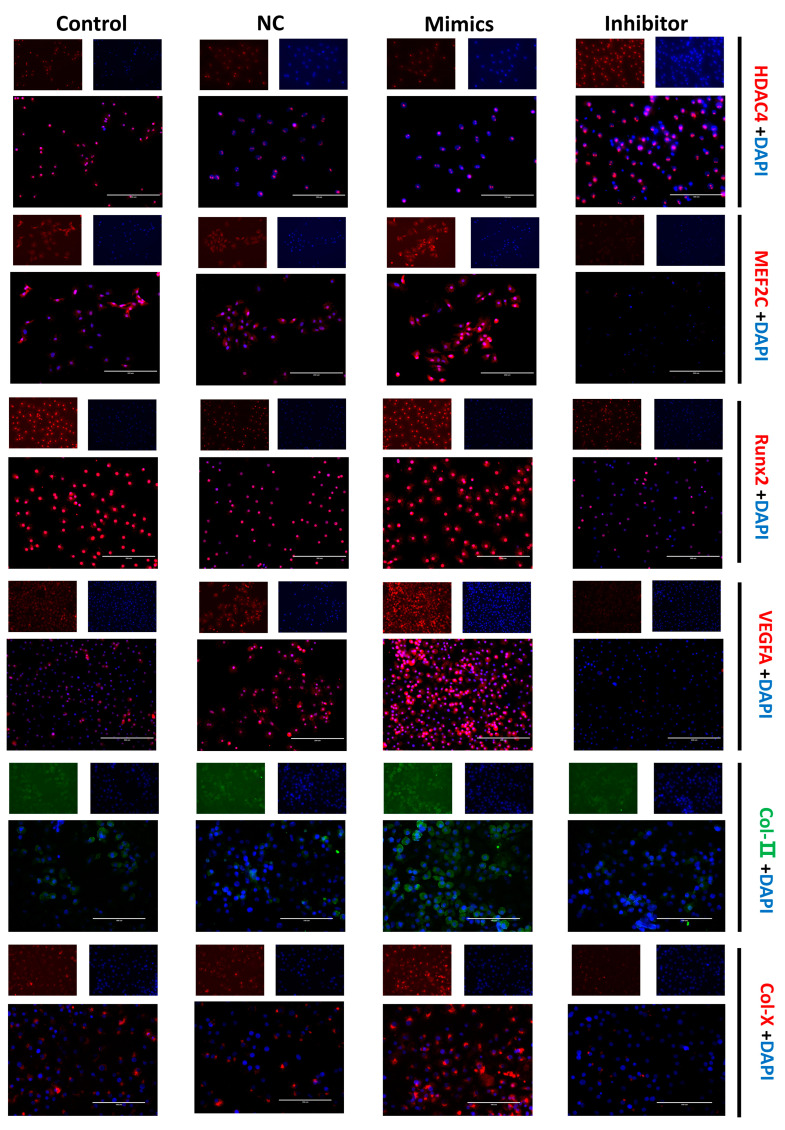
Effects of miR-140-5p on chondrocyte growth and differentiation marker protein expression and localization. Positive signals for protein expression are shown by red and green fluorescence, whereas blue represents the nucleus. The scale bars represent a length of 200 µm. The fluorescence signal of the HDAC4 protein was stronger in the miR-140-5p inhibitor group than in the control group. In the inhibitor group, fluorescent signals from proteins such as MEF2C, Runx2, VEGFA, Col-II, and Col-X were scarcely discernible. However, in the miR-140-5p mimics group, these proteins’ fluorescence signals were clearly evident, and suggesting increased expression.

**Figure 6 ijms-24-10975-f006:**
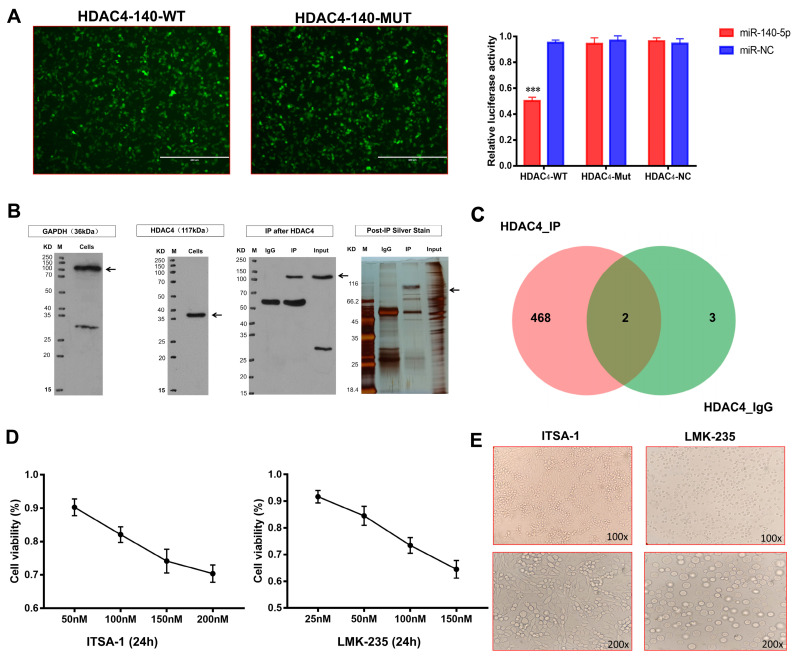
Mechanism of miR-140-5p in coordinating growth and differentiation of growth plate chondrocytes. (**A**): Validation of the targeting relationship between miR-140-5p and HDAC4 by dual-luciferase reporter gene assay. Cells with green fluorescence are 239T cells successfully co-transfected with HDAC4-WT, MUT, and gga-miR-140-5p mimics (scale bars, 400 μm). (**B**): Immunoblotting of HDAC4 interacting proteins screened by co-immunoprecipitation. Black arrows represent target protein bands. (**C**): Venn diagram of differential protein set between HDAC4_IP and HDAC4_IgG. (**D**): Evaluation of optimal concentrations of HDAC4 agonist ITSA-1 and HDAC4 inhibitor LMK-235 by CCK-8 cytotoxicity assay. (**E**): External morphological analysis of chondrocytes in each group after 24 h of administration (upper is 100X, down is 200X). “*” shows the level of significance among groups. *** *p* < 0.001.

**Figure 7 ijms-24-10975-f007:**
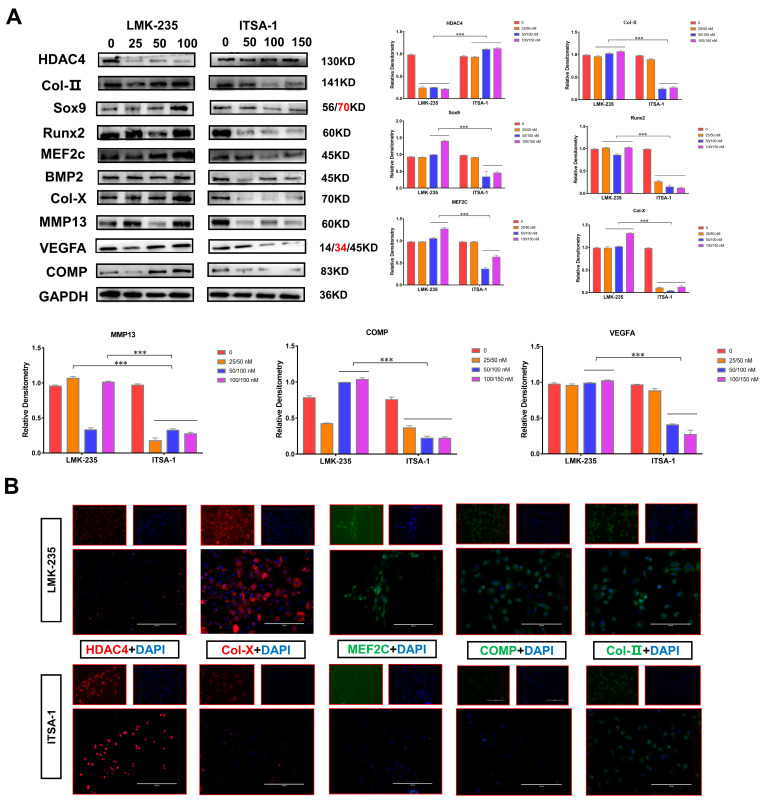
(**A**): Expression of chondrocyte growth and differentiation-related marker proteins after treatment with ITSA-1 and LMK-235 for 24h, using GAPDH as an internal reference protein. The densitometry of protein bands on the Fusion Solo S instrument. (**B**): A fluorescence experiment was used to examine the effects of ITSA-1 and LMK-235 on the expression and localization of chondrocyte growth and differentiation marker proteins. Red and green fluorescence were utilized as positive signals in this experiment to indicate protein expression, whereas blue fluorescence was used to observe the cell nucleus. A fluorescence microscope was used to view and record the fluorescence signals, and pictures were collected using a scale bar of 200 µm to give a reference for size measurements. The assay revealed a weak fluorescence signal of HDAC4 in the chondrocytes of the LMK-235 group. However, the fluorescence signals of Col-X, MEF2C, COMP, and Col-II were remarkably high, indicating their increased expression in response to treatment. The red text refers to a distinct band corresponding to targeted molecular weight (MW). “*” shows the level of significance among groups. *** *p* < 0.001.

**Figure 8 ijms-24-10975-f008:**
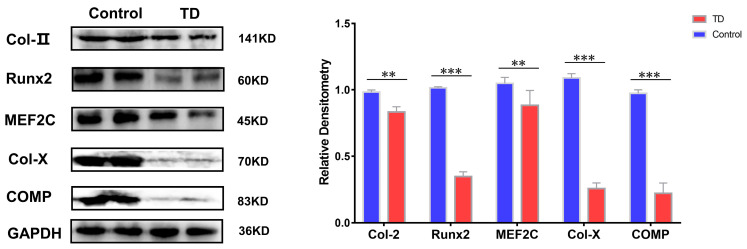
Expression verification of HDAC4 interacting proteins in animal models, GAPDH as internal reference protein. Repeated five times for each group. (** *p* < 0.01, *** *p* < 0.001).

**Figure 9 ijms-24-10975-f009:**
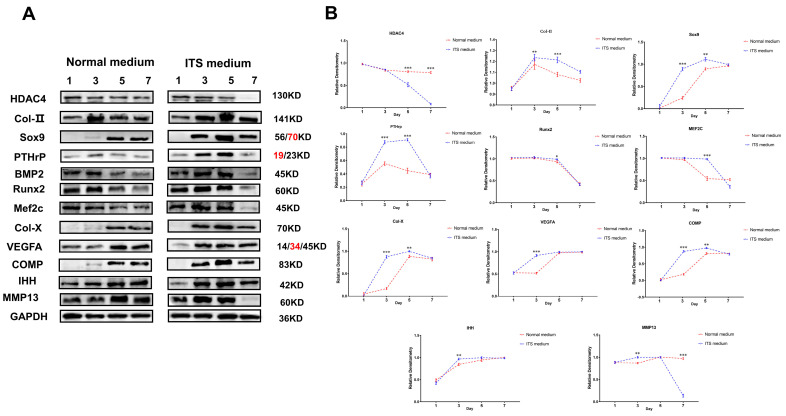
Dynamic monitoring of chondrocyte growth and differentiation marker proteins during chondrocyte differentiation. (**A**,**B**): Levels of chondrocyte growth and differentiation marker proteins in normal and differentiated states, using GAPDH as internal reference protein. Repeated five times for each group. The red text refers to a distinct band corresponding to targeted molecular weight (MW). (** *p* < 0.01, *** *p* < 0.001).

**Figure 10 ijms-24-10975-f010:**
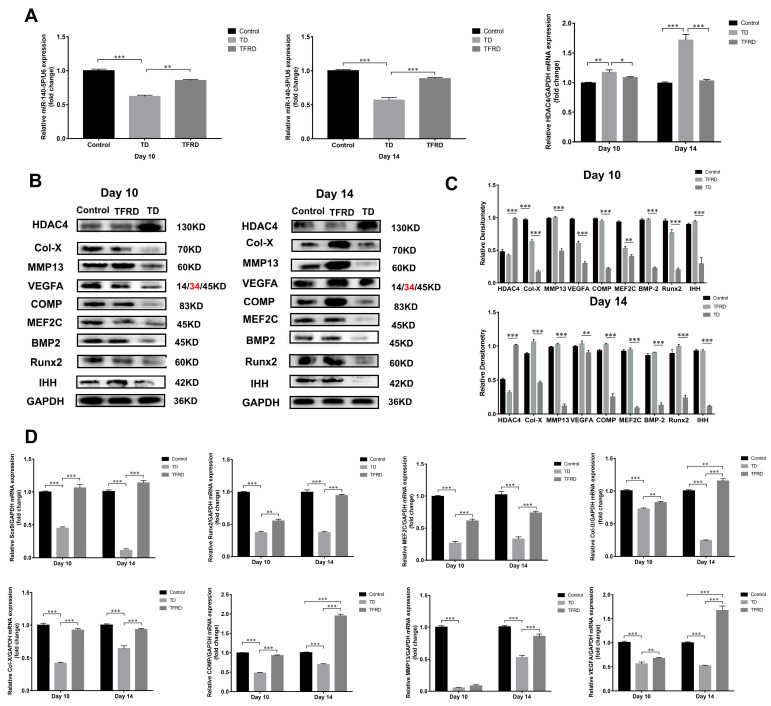
Effects of total flavonoids of *Rhizoma Drynariae* on the expression of miR-140-5p, HDAC4, and chondrocyte growth differentiation markers. (**A**): Expression of miR-140-5p and HDAC4 mRNA in the growth plate of each group on the 10th and 14th days using U6 as the housekeeping gene for miRNA and GAPDH as the housekeeping gene for mRNA. (**B**,**C**): Expression of growth and differentiation marker proteins of chondrocytes in the growth plate of each group on the 10th and 14th day using GAPDH as an internal reference protein. (**D**): Expression of growth and differentiation marker mRNA of chondrocytes in the growth plate of each group on the 10th and 14th days using GAPDH as a housekeeping gene. Repeated five times for each group. The red text refers to a distinct band corresponding to targeted molecular weight (MW). (* *p* < 0.05, ** *p* < 0.01, *** *p* < 0.001).

**Figure 11 ijms-24-10975-f011:**
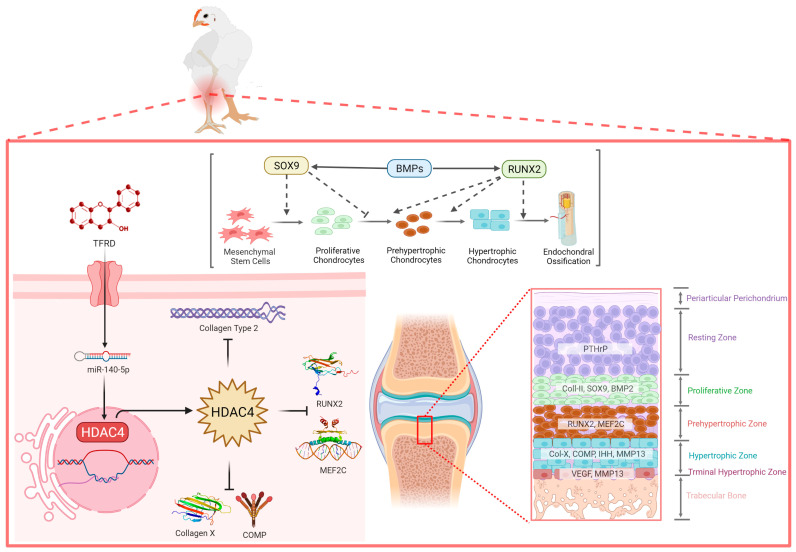
Understanding the complex interaction between miR-140-5p and HDAC4 in the growth plate to control the chondrocyte proliferation, differentiation, and maturation.

## Data Availability

Data are contained within the article.

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
