# Peer review of "The Effect of miR-140-5p with HDAC4 towards Growth and Differentiation Signaling of Chondrocytes in Thiram-Induced Tibial Dyschondroplasia"

_ijms, 2023, doi:10.3390/ijms241310975_

Round 1

Reviewer 1 Report

The Abstract should contain more info on the study design and focus. Were animals randomized? Lineages controlled? How generalizable are these results? Did you attempt any kind of validation that would confirm the findings? Which statistical software was used? How were the three measurements that you performed agreed in their numerical output, can you show some estimate of agreement, like kappa, ICC or even a CV? You should not report P as <0.01, but provide the exact P value; only for very low P values you can use this, P<0.001. TFRD is not mentioned in the Abstract, and it seems relevant to the study.   

Reasonable

Author Response

Reviewer 1:

Q: The Abstract should contain more info on the study design and focus. Were animals randomized? Lineages controlled? How generalizable are these results? Did you attempt any kind of validation that would confirm the findings? Which statistical software was used? How were the three measurements that you performed agreed in their numerical output, can you show some estimate of agreement, like kappa, ICC or even a CV? You should not report P as <0.01, but provide the exact P value; only for very low P values you can use this, P<0.001. TFRD is not mentioned in the Abstract, and it seems relevant to the study.

Response: We sincerely appreciate your valuable feedback and insightful questions. Allow us to address each of your concerns below:

  • By keeping your point of concern, we have updated the abstract and added more information regarding study design and focus.
  • Regarding the randomization of animals and control of lineages, we apologize for not providing sufficient information in our manuscript. In this study, animals were indeed randomized into different experimental groups to minimize bias and ensure equal distribution of potential confounding factors. Additionally, we took measures to control lineages by carefully selecting animals from the same breed, age, and sex for each experimental group. These details have already been included in the revised manuscript to address your concerns.
  • We recognize the significance of validating our findings to strengthen the robustness and reliability of our results. In this study, we conducted multiple experiments and employed various techniques to support our observations in order to confirm our findings and provide additional evidence for the observed effects.
  • We utilized GraphPad Prism 7 program for conducting statistical analyses. The specific version used (8.0.2) has been mentioned in the revised manuscript to provide transparency and reproducibility. Moreover, we acknowledge the importance of assessing the agreement between measurements as a measure of reliability. To properly assess the agreement between the three measurements, we understand the need for a quantitative measure. However, in the current study, we regrettably did not incorporate these specific agreement measures during our data analysis. Because in our study, the primary focus was on investigating the relationship between the measured variables and evaluating statistical significance between different groups. The three measurements performed were intended to capture different aspects or parameters related to our research question rather than assessing the agreement between them. As a result, our initial analysis approach did not include specific agreement measures such as kappa, intraclass correlation coefficient (ICC), or coefficient of variation (CV).
  • Regarding to write exact probability values, the our focus is primarily on evaluating statistical significance rather than providing precise numerical values for the P values. The use of "<0.01" is a widely accepted convention in scientific literature to indicate a significant result without the need for excessive decimal places. We believe that presenting the exact P values in this context may not provide meaningful additional information to the readers and could potentially clutter the text with an abundance of numerical values. By using the "<0.01" notation, we aim to convey the significant findings succinctly and efficiently.
  • Nonetheless, we understand your concern and appreciate your perspective on this matter. We will carefully reconsider your suggestion and evaluate its potential impact on the clarity and readability of our manuscript.
  • Thank you once again for your valuable input, and we will ensure that our revised manuscript addresses your concerns to the best of our abilities.
  • We appreciate the reviewer's comment regarding the relevance of TFRD to our study. However, we apologize for the oversight in not including TFRD in the abstract. The omission of TFRD in the abstract was a mistake, as it is indeed an important aspect of our research. We have taken note of this feedback and ensured that the abstract accurately reflects the inclusion of TFRD in our study.

Thank you for your valuable feedback and suggestions regarding our manuscript. Your insightful suggestions have provided us with valuable guidance and have helped us improve the clarity and scientific rigor of our study. We genuinely appreciate your expertise and the attention to detail you have demonstrated throughout the review process. Once again, we extend our sincere gratitude for your time and efforts in reviewing our work. Should you have any further suggestions or comments, we would be more than happy to address them.

Reviewer 2 Report

The paper is interesting and well written, but it can be improved:

- Figure 1 and related results: some parameters characterizing the condrocytes in the growth plate should be quatified; thus, column height and number of cells for hypertrophic and proliferative zone should be provided;

- Different graphs should be improved, it is very difficult to read them, i.e Figure 2B, 2C, 3B, 4, 7A, 9B, 10;

- Cell microphotography sholud be improved, with particular regard to that with fluorescence stain. These last require also a quantification of the positivity.

Author Response

Reviewer 2:

The paper is interesting and well written, but it can be improved:

  • Figure 1 and related results: some parameters characterizing the chondrocytes in the growth plate should be quantified; thus, column height and number of cells for hypertrophic and proliferative zone should be provided.

Response: Thank you for your valuable feedback and suggestions. We agree with your suggestion for providing quantification of column height and the number of cells for the hypertrophic and proliferative zones regarding chondrocyte characteristics in the growth plate. In response to your recommendation, we carefully analyzed our data and performed the necessary measurements for cells in the hypertrophic and proliferative zones. We have incorporated this quantification into the supplementary file of our revised manuscript, providing a more comprehensive description of the chondrocytes. Also, we have updated the Figure. 1 according to your concerns. We are hoping that it will cover up your suggested points.

  • Different graphs should be improved, it is very difficult to read them, i.e Figure 2B, 2C, 3B, 4, 7A, 9B, 10.

Response: Thank you for your feedback regarding the readability of graphs in our manuscript. We apologize for any difficulties encountered while interpreting these figures. To address this concern and ensure better understanding of the data, we have uploaded the original high-resolution figures to the submission system. These figures provide enhanced clarity and resolution compared to the versions initially included in the manuscript. We believe that by accessing the original figures, the readers will have a clearer visualization of the data and improved readability. Moreover, we have uploaded figures in Journal’s submission system in high-resolution original figures, thereby ensuring that readers have a clear representation of the data. We appreciate your understanding in this matter.

  • Cell microphotography should be improved, with particular regard to that with fluorescence stain. These last require also a quantification of the positivity.

Response: We appreciate your feedback regarding the cell microphotography, particularly those involving fluorescence staining, in our manuscript. Actually, our experimental design was primarily oriented towards understanding the molecular mechanisms and functional aspects of miR-140 and HDAC4 in chondrocytes. Therefore, our study did not specifically aim to quantify the positivity of fluorescence staining. Moreover, here it is worth mentioning that the conducting a comprehensive quantification of positivity, particularly for fluorescence staining, typically requires additional resources, including specialized image analysis software, time-intensive manual scoring, and trained personnel. Unfortunately, due to resource limitations, we were unable to allocate the necessary time and equipment to perform such a quantification. However, we assure you that we have taken great care to present the cell microphotography in a clear and informative manner. In the revised manuscript, we have enhanced the figure quality and provide detailed figure legends to ensure that readers can comprehend the observed features and differences between experimental groups.

Thank you for your valuable feedback and suggestions regarding our manuscript. Your insightful suggestions have provided us with valuable guidance and have helped us improve the clarity and scientific rigor of our study. We genuinely appreciate your expertise and the attention to detail you have demonstrated throughout the review process. Once again, we extend our sincere gratitude for your time and efforts in reviewing our work. Should you have any further suggestions or comments, we would be more than happy to address them.

Round 2

Reviewer 2 Report

The authors addressed all my concerns.